# Principles of Palliative and Supportive Care in Pancreatic Cancer: A Review

**DOI:** 10.3390/biomedicines11102690

**Published:** 2023-10-01

**Authors:** Robert Mazur, Jan Trna

**Affiliations:** Department of Gastroenterology and Digestive Endoscopy, Masaryk Memorial Cancer Institute, Zluty Kopec 7, 656 53 Brno, Czech Republic; robert.mazur@mou.cz

**Keywords:** pancreatic adenocarcinoma, pancreatic cancer, palliative care, supportive care

## Abstract

Pancreatic adenocarcinoma (PDAC) is well known for its poor survival time. Clinical symptoms are painless jaundice or abdominal or back pain. Less specific symptoms often appear that make diagnosis difficult, e.g., weight loss, loss of appetite, nausea and vomiting, and general weakness. Only 10–20% of patients are diagnosed at an early stage. A cure is practically only possible with a radical surgical operation. In the case of locally advanced findings, neoadjuvant therapy is administered. Among the therapeutic options offered are chemotherapy, radiotherapy (including stereotactic radiotherapy—SBRT), targeted treatment, or immunotherapy. In the case of metastatic disease, of which more than half are present at diagnosis, the goal is to relieve the patient of problems. Metastatic PDAC can cause problems arising from the localization of distant metastases, but it also locally affects the organs it infiltrates. In our review article, we focus on the largest group of patients, those with locally advanced disease and metastatic disease—symptoms related to the infiltration or destruction of the pancreatic parenchyma and the growth of the tumor into the surrounding. Therefore, we deal with biliary or duodenal obstruction, gastric outlet syndrome, bleeding and thromboembolic diseases, pain, depression, and fatigue, as well as pancreatic exocrine insufficiency and malnutrition. Metastatic spread is most often to the liver, peritoneum, or lungs. The presented overview aims to offer current therapeutic options across disciplines. In accordance with modern oncology, a multidisciplinary approach with a procedure tailored to the specific patient remains the gold standard.

## 1. Introduction

Pancreatic adenocarcinoma (PDAC) is a malignant tumor originating from the exocrine cells of the pancreas. It is most often adenocarcinoma (up to 95 percent), and the rest are cystic tumors or other rare types [1]. The malignant cells themselves originate from the epithelium of the pancreatic ducts. Risk factors include age (peak incidence in the 7–8th decade); smoking; excessive alcohol intake; or dietary factors such as a higher intake of meat, cholesterol, and fried foods [2]. Patients with a hereditary burden (p53 mutation, Lynch syndrome, BRCA 2 mutation), hereditary pancreatitis (trypsinogen gene defect), chronic pancreatitis, or type 2 diabetes are also at higher risk. PDAC, like most tumors, arises through a multi-stage process where the cumulative effect of individual mutations in individual genes, gene amplification, structural rearrangements, and deletion or loss of heterozygosity apply. The most frequent mutations occur in proto-oncogenes (K-ras, HER-2/neu) or tumor suppressor genes (p16, p53, BRCA 2, DPC4/SMAD4) [3]. On the basis of these genetic changes, risk lesions with malignant potential are described (PanIN (pancreatic intraepithelial neoplasia), together with IPMN (intraductal papillary mucinous neoplasms) and MCN (mucinous cystic neoplasms) [4]. Pancreatic cancer can be clinically asymptomatic until late stages or have non-specific symptoms such as fatigue, weight loss, or loss of appetite. Possible symptoms are based on the localization of the tumor—obstructive icterus when present in the head of the pancreas, vomiting in the case of duodenal oppression, and pain in the epigastrium with propagation to the back. At an advanced stage, stiff resistance in the epigastrium, ascites, or shortness of breath with metastatic involvement of the lungs may appear as the first symptom [5]. As part of the diagnosis, it is possible to find a pancreatic tumor using abdominal ultrasound, abdominal contrast CT, or endosonography (EUS). Many clinical studies are trying to determine the ideal marker with which it would be possible to diagnose tumors at an early stage [6]. Circulating tumor cells in the peripheral blood or specific molecules in the stool are looked for. Finding these molecules could lead to a breakthrough in the care of PDAC patients. In recent years, there has been an enormous effort in the field of scientific research to find a suitable methodology for effective screening [2,5].

So far, PDAC is still at the lower end of the survival rate despite great advances in medicine in recent decades. The incidence of PDAC is increasing, and it is projected to become the second-leading cause of cancer-related mortality by 2030. Most patients are diagnosed with locally advanced (30–35%) or metastatic (50–55%) disease [7]. Currently, the five-year survival rate is less than 10%. Surgical treatment is the only potentially curative treatment if the tumor is identified at a resectable stage. Due to non-specific or non-existent symptoms in the early stages, we often discover tumors suitable for surgery accidentally during the examination of patients rather for other reasons [8]. Due to the large number of patients with metastatic pancreatic cancer and locally advanced PDAC, in this review article, we focus on the most common symptoms and their treatment options according to current recommendations. According to NCCN guidelines, the main symptoms encountered in clinical practice are biliary obstruction, gastric outlet/duodenal obstruction, thromboembolic disease, bleeding from the primary tumor, pain, depression and fatigue, and exocrine pancreatic insufficiency and malnutrition, with symptoms resulting from the local growth of the tumor, its metastasis (most often to the liver, peritoneum, and lungs), and the production of various molecules [9,10].

## 2. Biliary Obstruction

Painless jaundice is perceived as one of the most common manifestations in patients with ductal adenocarcinoma of the pancreatic head and occurs in approximately three-quarters of this population [9]. Obstruction is primarily caused by the growing tumor compressing the bile ducts from the outside or by endoluminal growth. Endoscopic retrograde cholangiopancreatography (ERCP) with the introduction of a self-expandable metallic stent (SEMS) is the gold standard for biliary drainage [11].. ERCP is preferred over percutaneous transhepatic biliary drainage (PTBC) due to longer survival rates, fewer infectious complications, and shorter hospitalizations [12]. A plastic stent should not be inserted due to the need for regular replacements, as well as more frequent complications such as blockage or cholangitis, so that a metallic stent is ultimately more advantageous despite the higher initial price [13]. In recent years, thanks to technological progress, the concept of EURCP (endoscopic ultrasound retrograde cholangiopancreatography) has been promoted in specialized centers. It was possible to connect the diagnostic accuracy of EUS (endoscopic ultrasound) with the therapeutic possibilities of ERCP. Previously, in the event of failure to ensure drainage of the bile ducts, it was necessary to use PTBC or proceed to surgery. Today, the benefits of EURCP can be exploited with success [14]. One possibility is the detection of dilated bile ducts via EUS and their puncture in a suitable place. A wire is introduced, which then passes through the major duodenal papilla. After visualization of the location of the bile duct, the bile ducts are opened by ERCP (the EUS-rendezvous technique—EUS RV) [15]. With gradual use in practice, several other options have been developed to ensure the drainage of the bile ducts (see Figure 1). If it is impossible to insert a guide wire through the major duodenal papilla, it is possible to drain the bile directly into the stomach antegradely. Another option is hepaticogastrostomy (HGS) or choledochoduodenostomy (CDS). When performing transmural stenting, a special stent—a lumen-apposing metal stent (LAMS)—has been designed for this purpose [16]. 

The previously often-used PTBC is gradually being replaced by EURCP. A systematic review and meta-analysis by Hayat et al. compared technical and clinical success rate and acute, delayed, and total adverse events of EUS-BD (biliary drainage) with PTBC. The result showed that EUS-BD is comparably effective but safer in terms of acute and overall adverse events for biliary decompression in patients with malignant biliary strictures for whom ERCP has failed [17]. It is also comparable to the effectiveness of surgical procedures and is increasingly used due to its minimal invasiveness [18,19].

## 3. Gastric Outlet Obstruction (GOO)

Gastric outlet obstruction of any degree is described in approximately one-sixth to one-fifth of patients [20], and one-tenth of all patients with pancreatic cancer develop symptomatic duodenal obstruction [21]. Patients are bothered by persistent nausea and pressure or pain in the epigastrium due to due to the inability of the contents from the stomach to move on. The main goal of therapy is the relief of symptoms and, if possible, the restoration of oral intake [22]. Symptoms can be influenced by diet modification or medication. Prokinetics, antiemetics, or analgesics have been used with partial success. In terms of restoring food intake naturally, we have a choice of endoscopic or surgical methods. In a good clinical condition, surgical gastrojejunostomy via either a laparoscopic or an open approach is possible. Compared to an enteric stent inserted endoscopically, this has a longer effect and a lower frequency of reinterventions, with the cost of longer hospitalization and the higher financial cost of the procedure [23]. Currently, with the increased availability of quality gastrointestinal endoscopy, the use of enteric stents is the preferred method (see Figure 2). It is possible to perform this in a shorter time horizon. Moreover, it leads to a faster relief, according to a study by Orr et al. who compared the effect of an enteric stent in patients with PDAC and other malignancies in terms of the need for reintervention. Only 1 patient out of 26 patients with PDAC had to be re-intervented, compared to 6 patients out of 16 with other malignancies (OR: 0.064, 95% CI: 0.01–0.60) [24]. Unfortunately, according to studies to date, neither method has an effect on life expectancy. However, after endoscopic stenting, some patients may be candidates for systemic therapy [25,26].

The third option, the already mentioned LAMS (lumen-apposing metallic stent), has also been used in these cases in recent years (see Figure 3). This new technique ideally combines the advantages of minimal invasiveness of an endoscopic procedure and the long-term effect of a surgical solution [27]. The new CABRIOLET study evaluated the most appropriate therapy for combined bile duct obstruction and gastric outlet obstruction. Five possible approaches, namely, enteral stenting (ES), EUS-guided gastroenterostomy (EUS-GE), hepaticogastrostomy (EUS-HGS), choledochoduodenostomy (EUS-CDS), and transpapillary biliary stenting (TPS), were assessed for dysfunction during follow-up. Although the authors assessed insufficient statistical evidence for individual combinations, it appears that the combination of EUS-GE+HGS or EUS-GE+TPS may lead to better patency [28].

## 4. Thromboembolic Disease

Cancer is an independent and major risk factor for venous thromboembolism (VTE), defined as deep-vein thrombosis (DVT), including catheter-related thrombosis (CRT) and pulmonary embolism (PE). PDAC carries the highest risk of VTE of all cancers, with VTE rates ranging from 5 to 41% in retrospective cohorts and up to 67% in postmortem series [29].

Detailed pathological mechanisms and relationships between individual pathways are still the subject of intensive studies [30]. Recommendation of the International Initiative on Thrombosis and Cancer agreed on the primary prevention of thromboembolism using LMWH (low-molecular-weight heparin) or DOAC (direct oral anticoagulants—in the case rivaroxaban or apixaban) in ambulatory patients at low risk of bleeding with metastatic pancreatic tumor on antitumor treatment [31]. In the case of a higher risk of bleeding, LMWH remains the standard of therapy. More recently, DOAC is also recommended for patients with VTE, especially in the absence of a tumor mass in the lumen of the digestive tract [32].

## 5. Bleeding from Primary Tumor Site

Bleeding into the gastrointestinal tract is a rare but serious complication of pancreatic adenocarcinoma, mainly due to difficult hemostasis. According to Takada et al., approximately 36% of all PDAC patients have duodenal infiltration, and 2.6% of PDAC patients manifest primarily with gastrointestinal bleeding [33]. The first and most affordable response is therapeutic endoscopy with several hemostatic options available. The most common are adrenaline injection, clipping, hemospray, or stent insertion, or a combination of these modalities. If this fails, TAE (transarterial embolization), surgery, and radiation are offered. TAE is more suitable for arterial bleeding. In the case of choosing a surgical solution, we must expect a possible increase in complications and a failure to improve the patient’s prognosis. Radiation, if not previously performed in the tumor area, can stop bleeding if the endoscopy fails [34].

## 6. Pain

Most PDAC patients have pain at the time of diagnosis, although the pain intensity varies depending on the PDAC location in the pancreas, the person’s anatomy, and the presence of metastatic lesions or local invasion [35].

Managing pain as a common symptom in advanced pancreatic cancer can be a challenge. When it is successfully managed, the quality of life improves. However, it often requires a multidisciplinary and sometimes invasive approach, which is not available in all healthcare institutions [36]. Early referral of the patient to a palliative specialist can improve tailoring of treatment [37].

Pain management modalities include medications, chemotherapy and radiotherapy, interventions, and alternative approaches. In the case of mild pain, it is advisable to proceed according to the analgesic ladder. In the case of severe pain, it is sometimes necessary to skip the ladder and proceed directly to opioids and their derivatives (see Table 1). Administration is possible orally, transdermally, sublingually, or intranasally [38]. Opioids must be used carefully in view of their considerable potential for side effects, particularly sedation and respiratory depression, itching, nausea, and constipation. Stool softeners and laxatives for constipation, diphenhydramine for itching, and antiemetics for nausea are recommended as prevention or relief [39]. Corticoids, antidepressants, or anticonvulsants are also used as adjuvant drugs to deal with side effects [40].

Pain control is largely implemented as a secondary objective in trials evaluating chemotherapyregimens in PDAC. Radiotherapy is particularly effective in controlling and relieving pain caused by large tumors pressing on other organs or structures, such as nerves or the spine. The effect of radiotherapy is usually late and appears many weeks after the start of treatment. Radiotherapy can shrink the tumor, which may help in relieving the pain [41]. An alternative approach is to use a celiac plexus block or neurolysis and can be performed minimally invasively under CT or endoscopically under EUS control or surgically [39,42,43]. Intrathecal therapy can be suggested for terminal stages or in refractory pain. The implantable intrathecal drug delivery systems are an alternative with reduced drug toxicity and improved pain scores. Various molecules can be used, including morphine, fentanyl, local anesthetics, baclofen, and/or clonidine [44]. Of the alternative approaches, a systemic review published in 2012, including a total of 15 randomized controlled trials, showed that acupuncture is an effective adjunctive method in the treatment of cancer pain, and relief from pain was better when compared to drug therapy alone [45]. Hypnosis is also one of the approaches used to control pain. Several studies confirm that it is useful in reducing cancer-related pain, modulating the sensation of pain by functional disconnection between the prefrontal cortex, the decision center and the anterior cingulate cortex [46].

**Table 1 biomedicines-11-02690-t001:** Recommended oral analgesics for treatment of PDAC according to Hameed et al. [40] and Drewes et al. [42].

Drug Class	Example	Comments
Non-opioid analgesics	paracetamol (acetaminophen) 1000 mg × 4, ibuprofen 200–400 mg × 4	Used for milder pain.
Weak opioids	tramadol CR 50–200 mg × 2, codeine 15–60 mg × 4	Potentiates the effect of non-opioid analgesics.
Strong opioids	oxycodone PR starting at 15 mg × 2, morphine starting at 30 mg × 3,	Some patients may develop opioid-induced hyperalgesia.
Anticonvulsants	pregabalin titrated from 75 mg to 300 mg × 2 daily	Side effects (mainly drowsiness and dizziness) vanish during treatment.
TCA	amitriptyline 10–50 mg at nighttime	The effects appear after several weeks of treatment. If one TCA does not work, another may be effective.
SSRI	citalopram titrated up to 40 mg at nighttime	May be preferred by some patients in case of comorbid anxiety and depression.
SNRI	duloxetine titrated up to 120 mg at nighttime	May be used when neuropathic pain is suspected, and for comorbid anxiety or depression.
Anxiolytics	diazepam 5 mg × 3	Mainly an anxiolytic effect.
Anti-psychotics	levomepromazane titrated up to 100 mg daily	May potentiate the analgesic effect.
Cannabinoids	nabilone dose 1–2 mg, up to 6 mg daily	Adjunctive pain medication and useful for nausea, appetite, and sleep.
Steroids	dexamethasone 8 mg daily	Anti-inflammatory effects in selected cases.
GABA-agonists	baclofen titrated from 5 to 25 mg × 3	May be effective in neuropathic pain.

## 7. Depression and Fatigue

Due to the severity of metastatic PDAC, depression and fatigue occur quite often. Their timely detection is therefore absolutely essential [47]. In practice, numerical analogue scales, a thermometer, or simple scoring systems are used to detect this. Patients, along with oncologists and other healthcare professionals, must balance between realistic expectations and life-limiting diagnosis [48].

The prevalence of depression has been shown to be up to seven times higher in PDAC patients than in the general population. A meta-analysis of six prospective studies calculated that up to 43% of patients were depressed after the diagnosis. Therefore, it is essential to seek out these patients and adequately help them from the beginning of the diagnosis. It should be borne in mind that depression can appear with an illness or at any time later when the patient is already being treated for it. The initial assessment depends on factors that could trigger it, such as the disease itself, the treatment or symptoms, or an adverse social environment. It is appropriate for the patient to be referred for formal assessment of the problem by the palliative care team. Therapy includes, in addition to optimal symptom management, psychotherapy, psychotropic drugs, and the involvement of a social worker or chaplain service. Attention should also be paid to close relatives of the patient, who also have a higher risk of depression, with the use of anxiolytics or the long-term use of hypnotics [49].

The non-pharmacological treatment of fatigue consists in physical activity, carefully optimized with regard to the patient’s goals and limited by the scope of the disease. It is necessary to consider whether the patient has bone metastases, thrombocytopenia, anemia, fever or active infection, or other limitations resulting from metastases or comorbidities [50]. Fatigue can also be influenced by medication. Methylphenidate, which is still under research, with exact doses and dosages not yet determined, and corticoids, namely, prednisone and dexamethasone, are considered in a short-term regimen [51].

## 8. Exocrine Pancreatic Insufficiency and Malnutrition

Pancreatic exocrine insufficiency (PEI) is one of the causes of malnutrition and weight loss in PDAC patients [52]. In general, the reason for PEI may be destruction of the pancreatic parenchyma by the tumor, obstruction of the pancreatic duct, fibrosis of the gland, or surgery associated with resection of the pancreas. PEI and malnutrition have a negative effect on the quality of life and the management of complications resulting from the disease or treatment, and they also shorten life expectancy [53]. Despite this, nutritional therapy is not everywhere an integrated part of a comprehensive approach [54]. A rapid evaluation and referral of the patient to a diet specialist is necessary.

The standard PEI therapy is pancreatic enzyme replacement therapy (PERT); a systematic review and meta-analysis showed that its use was associated with a 3.8 month increase in survival (95% confidence interval: 1.37–6.19), and patients had a trend towards a better quality of life [52]. The dosage is in accordance with the European recommendations for chronic pancreatitis and recent review on PERT in PDAC [55,56]. Roeyen et al. recommend considering 40,000–50,000 units of lipase to be taken with each meal and 25,000 units to be taken with each snack as a suitable starting dose. It is also appropriate to administer PERT to achieve optimal nutritional status in all patients who have received neoadjuvant or adjuvant therapy or palliative therapy. The effectiveness of PERT is evaluated based on the alleviation of maldigestion-related symptoms (e.g., steatorrhea, weight loss, flatulence) and the normalization of the patients‘ nutritional status [57]. Proton pump inhibitors can be given as adjuvant therapy to prevent inactivation of PERT by gastric acids and thus improve the response [58]. PDAC patients are also more prone to small intestine bacterial overgrowth (SIBO) than healthy individuals. SIBO can aggravate the symptoms of digestive discomfort through multiple factors that affect the small intestine [59]. *Rifaximin*, a non-absorbable broad-spectrum antibiotic, may bring symptoms relief via restoration of healthy bowel microbiota [60].

Regarding parenteral nutrition, this should be considered in the case of expected survival longer than 2–3 months, whenever severe nausea, vomiting, diarrhea, constipation, stomach pain, or gastrointestinal obstruction do not allow enteral feeding and at the same time the usual procedures described in the previous text do not bring any improvement in status [61].

## 9. Metastatic Pancreatic Cancer

PDAC has a tendency to early progression to metastatic disease [62]. It is often able to metastasize, even in the case of a small primary tumor. In the case of metastatic disease, the liver is affected in up to 76–80% of patients, the peritoneum in 48%, and the lungs in 45% [10]. This order was confirmed in the autopsy findings [63]. The tumor can spread by direct growth, hematogenously or lymphogenously. Surgical resection remains the only effective method for cure and long-term survival, although few patients can undergo it [64]. Despite all efforts, the average survival time of metastatic PDAC is 3–6 months, with a 5 year survival rate of 1–2% [65]. According to Liu et al., the median overall survival (OS) time was 6.5 months in the entire cohort, while the median OS times were 11.8, 6.9, 7.7, 10.1, and 5.0 months in patients with isolated lung, isolated liver, isolated peritoneum, isolated distant lymph nodes, and metastases at multiple sites, respectively [66].

Palliative chemotherapy prolongs the survival of PDAC survival when compared with best supportive care (BSC). While BSC patients have a median survival of around 2 months, gemcitabine-treated patients have a median survival of 5–6 months and PDAC patients treated with modern chemotherapy regimens such as FOLFIRINOX or gemcitabine/nab-paclitaxel have a median survival of 8–11 months [67,68,69]. However, the overall advantage is uncertain according to some researchers, who question if the benefits of palliative chemotherapy in PDAC patients outweigh their risks. This uncertainty is further highlighted when considering a second line of chemotherapy and, thus, BSC would be an appropriate alternative in these cases [70]. Current research is trying to identify the groups of PDAC patients who may benefit from subsequent lines of chemotherapy regimens [71].

Liver metastases present with symptoms such as pain, weight loss, epigastric discomfort, jaundice, nausea and vomiting, anorexia, and ascites. Patients with good performance status are usually indicated for multimodal treatment including chemotherapy, targeted therapy, or immunotherapy [72]. In the case of progression to the peritoneum, patients often suffer from abdominal pain, intestinal motility disorder, ascites, and intestinal obstruction. The use of intraperitoneal chemotherapy or cytoreductive surgery did not result in a significant improvement in survival or quality of life [73]. Various symptomatic therapies are often the only option. Glucocorticoids, antiemetics, analgesics, and antisecretory drugs (anticholinergics, somatostatin analogs, and proton pump inhibitors) are used. A venting gastrostomy is appropriate in the case of medication failure. Every patient needs adequate hydration. Parenteral nutrition and pain treatment should be based on the current condition and needs of the specific patient [74].

Patients with lung metastases usually have persistent cough, dyspnea, hemoptysis, and recurrent lung infections. Symptomatic non-pharmacological (cough suppression) and pharmacological treatment (demulcents, opioids, peripherally acting antitussives, or local anesthetics), as well as endobronchial brachytherapy, often relieve patients of these problems [75]. For patients with malignant pleural effusion, talc pneumodesis or indwelling pleural catheter is available. Both therapeutic procedures are considered to be equally effective in terms of dyspnea relief, post-interventional quality of life, and complication rates. Talc pnemodesis results in a higher rate of successful pleurodesis than indwelling pleural catheter (relative risk: 1.56; 95% confidence interval: [1.26; 1.92]; *p* < 0.0001) according to Hoffmann et al. [76]. Dyspnea management can be difficult, and non-pharmacological interventions may be offered by use of a face-directed ventilator, oxygen supplementation (in case of hypoxemia), a therapeutic trial with high-flow nasal oxygen therapy, or non-invasive ventilation. Others include various breathing techniques, positioning, relaxation, meditation, self-management, physical therapy, and music therapy. Acupressure or reflexology may also be provided if available. The simultaneous use of non-pharmacological and pharmacological therapy is recommended. Of the many pharmacological options, we commonly use opiates, short-acting benzodiazepines, systemic corticoids, and bronchodilators (see Table 2). There is insufficient evidence for antidepressants, neuroleptics, or inhaled furosemide, but they are used in clinical practice. Refractory dyspnea and life expectancy of days may lead to the offer of continuous palliative sedation [77].

## 10. Conclusions

In recent years, unfortunately, there has been no fundamental change in PDAC therapy that would lead to a higher number of cured patients. Despite promising screening programs, the ideal tool for detecting early stages is still lacking. Slightly longer survival times and increasing incidence lead to more experience with advanced or metastatic PDAC. The growth of randomized trials in advanced PDAC is improving the care of these patients. An understanding of the molecular processes has led to recommendations for thromboembolic disease profiling. Technological progress in endoscopy has developed new methods of solving biliary or duodenal obstruction. The improved availability of palliative and supportive care has enabled the better control of pain, depression, and malnutrition, thus contributing to a better quality of life. Our review attempted to provide an overview of the current procedures and recommendations for solving problems and thus can help the experts who care for these patients in their daily practice.

## Figures and Tables

**Figure 1 biomedicines-11-02690-f001:**
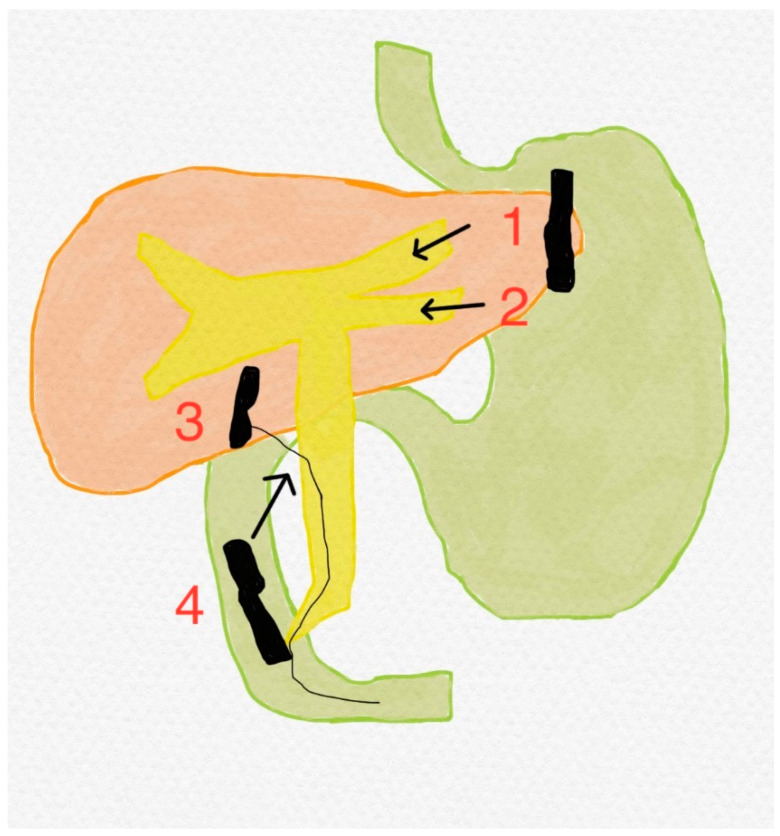
EUS-guided biliary drainage methods: (1) antegrade stenting; (2) and (4) transmural stenting with two kinds of techniques: (2) hepaticogastrostomy (preferred bile duct segment 3 or left liver) and (4) choledochoduodenostomy; (3) the rendezvous technique [16].

**Figure 2 biomedicines-11-02690-f002:**
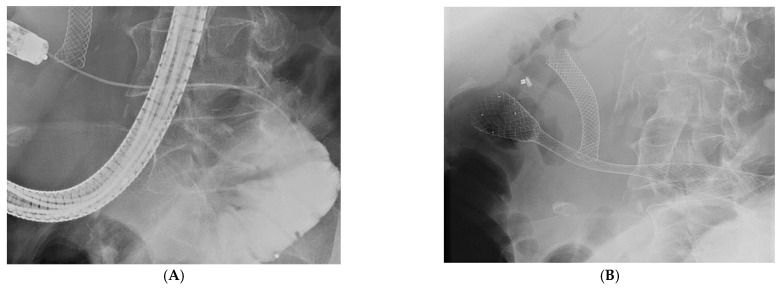
Enteral stent. (**A**) Endoscope with guidewire through the stenosis; (**B**) stent in the correct position (Author: Prof. Tomas Hucl, Department of Gastroenterology and Hepatology, Institute for Clinical and Experimental Medicine, Prague, Czech Republic).

**Figure 3 biomedicines-11-02690-f003:**
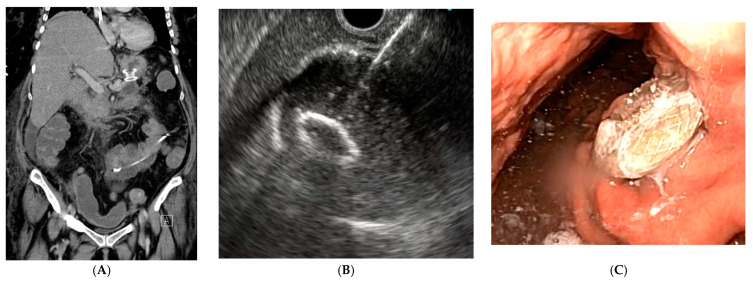
LAMS. (**A**) CT image; (**B**) EUS picture of LAMS opening; (**C**) endoscopic view (Author: Prof. Tomas Hucl, Department of Gastroenterology and Hepatology, Institute for Clinical and Experimental Medicine, Prague, Czech Republic).

**Table 2 biomedicines-11-02690-t002:** Pharmacological options of metastatic PDAC treatment (According to [78,79].

Medication	Dosage	Available Routes
Morphine	1–5 mg/4 h	p.o., s.c., i.v.
Hydromorphone	0.2–1.3 mg/4 h	p.o., s.c.
Lorazepam	0.5–1.0 mg/6–8 h	p.o., s.l.
Midazolam	2.5–5 mg/4 h	s.c., i.v.
Prednison	5–40 mg/day in 1–2 doses	p.o.
Dexamethasone	2–8 mg/day in 1–2 doses	p.o., i.m., i.v., s.c.

p.o. oral, s.c. subcutaneous, s.l. sublingual, i.m. intramuscular, i.v. intravenous.

## Data Availability

Not applicable.

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
