# Peer review of "Principles of Palliative and Supportive Care in Pancreatic Cancer: A Review"

_biomedicines, 2023, doi:10.3390/biomedicines11102690_

Round 1

Reviewer 1 Report

The authors provide a general overview of the management of symptoms and complications of locally progressed or metastatic Pancreatic adenocarcinoma (PDAC), representing the majority of cases in which PDAC is diagnosed. The authors do not describe which databases they examined and with which search strategy. This referent is a retired general physician with a recent case of locally advanced PDAC in his immediate family. He does not consider himself competent to verify the accuracy and coverage of the descriptions. He considers himself a person who will want to read a review such as this to inform himself globally.

The article lacks a representation of pharmacotherapeutic and surgical treatment options for primary tumor and metastases. Currently, successes are being achieved with the combination of Oxaliplatin, Irinotecan and 5FU (FOLFIRINOX). Also, in some parts of the world, the so-called Wipple surgery is performed laparoscopically and this referent expects this practice to expand. Also, the development of diabetes mellitus due to treatment (or otherwise) is not mentioned. 

The large number of abbreviations sometimes makes the article difficult for outsiders to follow. It is recommended that at least in each new paragraph an additional designation be used e.g. pancreatic carcinoma instead of PDAC. Also, it might be advisable to explain in a few words what is actually done in the various surgical procedures (indicating the anatomical localizations). The figure is not sufficiently clear in this regard.

Is giving pancreatic enzyme replacement therapy (PERT) the only option to treat/prevent malnutrition and enteral symptoms or should other supportive therapy be given? For example, what about the change of intestinal flora?

Author Response

1. comment:

The article lacks a representation of pharmacotherapeutic and surgical treatment options for primary tumor and metastases. Currently, successes are being achieved with the combination of Oxaliplatin, Irinotecan and 5FU (FOLFIRINOX). Also, in some parts of the world, the so-called Wipple surgery is performed laparoscopically and this referent expects this practice to expand. Also, the development of diabetes mellitus due to treatment (or otherwise) is not mentioned.

Answer: 

Topic of this review is not to describe exact chemotherapy regimens or surgical procedures, that are done in case of curative treatment intend. The topic is palliative and supportive care, long description of chemotherapy and/or surgery would make article more difficult to read and understand. However, paragraph about effect of palliative chemotherapy was added.

2. comment

The large number of abbreviations sometimes makes the article difficult for outsiders to follow. It is recommended that at least in each new paragraph an additional designation be used e.g. pancreatic carcinoma instead of PDAC. Also, it might be advisable to explain in a few words what is actually done in the various surgical procedures (indicating the anatomical localizations). The figure is not sufficiently clear in this regard.

Answer: 

Each abbreviation was explained in full when appearing for the first time. Palliative endoscopic drainage procedures were described in text.

3. comment

Is giving pancreatic enzyme replacement therapy (PERT) the only option to treat/prevent malnutrition and enteral symptoms or should other supportive therapy be given? For example, what about the change of intestinal flora?

Answer: paragraph about SIBO and its treatment with rifaximin was added. 

Reviewer 2 Report

Please add the difference in survival prognosis between patients with unresectable pancreatic cancer who were able to receive chemotherapy but chose BSC and patients with unresectable pancreatic cancer who were unable to receive chemotherapy and became BSC.

Minor editing of English language required

Author Response

Comment: 

Please add the difference in survival prognosis between patients with unresectable pancreatic cancer who were able to receive chemotherapy but chose BSC and patients with unresectable pancreatic cancer who were unable to receive chemotherapy and became BSC.

Answer:

Paragraph regarding CHT vs BSC was added alomg with several citations.

Round 2

Reviewer 1 Report

I thank the authors for their careful handling of the reviewers' suggestions. I have no substantive reservations about publication of the manuscript, but again point out that I am not particularly trained in this area of medicine. I noticed that during revision, spaces and sometimes punctuation were lost in several places in the text. Furthermore, the sentence in lines 234-235 was unclear to me: "Attention should also be paid to close relatives of the patient, who also have a higher risk of depression, with the use of anxiolytics or long-term use of hypnotics." Is it meant that particular restraint should be exercised with sleep medications and anxiolytics among close relatives, or rather that their prescription should be considered?

I noticed that during revision, spaces and sometimes punctuation were lost in several places in the text. 

Author Response

I noticed that during revision, spaces and sometimes punctuation were lost in several places in the text. 

Answer: this should be fixed during the professional preparation of the manuscript. Mistakes are result of track changes text processing. 

Furthermore, the sentence in lines 234-235 was unclear to me: "Attention should also be paid to close relatives of the patient, who also have a higher risk of depression, with the use of anxiolytics or long-term use of hypnotics." Is it meant that particular restraint should be exercised with sleep medications and anxiolytics among close relatives, or rather that their prescription should be considered?

Answer: yes, this sentense is not clear. It should say: Attention should also be paid to close relatives of the patient, who also have a higher risk of depression with a posibility of anxiolytics and/or hypnotics abuse. 
